# General Nutritional Profile of Bee Products and Their Potential Antiviral Properties against Mammalian Viruses

**DOI:** 10.3390/nu14173579

**Published:** 2022-08-30

**Authors:** Syeda Tasmia Asma, Otilia Bobiş, Victoriţa Bonta, Ulas Acaroz, Syed Rizwan Ali Shah, Fatih Ramazan Istanbullugil, Damla Arslan-Acaroz

**Affiliations:** 1Department of Food Hygiene and Technology, Faculty of Veterinary Medicine, Afyon Kocatepe University, Afyonkarahisar 03200, Turkey; 2Department of Beekeeping and Sericulture, University of Agricultural Sciences and Veterinary Medicine of Cluj-Napoca, 400372 Cluj-Napoca, Romania; 3Department of Animal Nutrition and Nutritional Diseases, Faculty of Veterinary Medicine, Afyon Kocatepe University, Afyonkarahisar 03200, Turkey; 4Department of Food Hygiene and Technology, Faculty of Veterinary Medicine, Kyrgyz-Turkish Manas University, Bishkek KG-720038, Kyrgyzstan; 5Department of Biochemistry, Faculty of Veterinary Medicine, Afyon Kocatepe University, Afyonkarahisar 03200, Turkey

**Keywords:** antiviral properties, nutrients, bioactive substances, honey, bee pollen, propolis, bee venom, royal jelly, bee bread, beeswax

## Abstract

Bee products have been extensively employed in traditional therapeutic practices to treat several diseases and microbial infections. Numerous bioactive components of bee products have exhibited several antibacterial, antifungal, antiviral, anticancer, antiprotozoal, hepatoprotective, and immunomodulatory properties. Apitherapy is a form of alternative medicine that uses the bioactive properties of bee products to prevent and/or treat different diseases. This review aims to provide an elaborated vision of the antiviral activities of bee products with recent advances in research. Since ancient times, bee products have been well known for their several medicinal properties. The antiviral and immunomodulatory effects of bee products and their bioactive components are emerging as a promising alternative therapy against several viral infections. Numerous studies have been performed, but many clinical trials should be conducted to evaluate the potential of apitherapy against pathogenic viruses. In that direction, here, we review and highlight the potential roles of bee products as apitherapeutics in combating numerous viral infections. Available studies validate the effectiveness of bee products in virus inhibition. With such significant antiviral potential, bee products and their bioactive components/extracts can be effectively employed as an alternative strategy to improve human health from individual to communal levels as well.

## 1. Introduction

Bee products offer propitious health benefits and are more and more popular in the era of medicinal research. They have also been well known for their nutritional benefits and therapeutic activity since approximately 5500 years ago [1]. In ancient times, people collected honeybees individually by hand, which is still in practice [2]. Several bee products (honey, propolis, bee pollen, bee bread, royal jelly, bee venom, and bee wax) have been recognized over time as potential sources of bioactive substances with medicinal potential in the treatment of several infections caused by viruses, bacteria, and parasites [3,4]. Moreover, viruses play an important role as causative agents in different cancer types, which are deadly diseases and the leading cause of worldwide deaths [5,6,7]. It was reported that viruses induce about 1.4 million cancers per year, which is roughly 10% of worldwide cancer cases [8]. Several studies have reported that commonly used bee products such as honey, propolis, royal jelly, bee pollen, and bee venom have significant potential for the treatment of different types of cancers [3,9].

Honey is an oxidizing agent with anti-inflammatory, antiproliferative, immunomodulatory, proapoptotic, antimetastatic, and antimicrobial properties [10,11,12]. Bee venom acts as apitoxin or biotoxin prepared by bee glands and secreted in the abdominal cavity of bees. It was also found to be effective in the treatment of different types of cancers, with effects such as cancer cell proliferation inhibition, induction of apoptosis, cytotoxicity, and necrosis [13]. Propolis, commonly known as bee glue, is made up of pollen, essential oils, waxes, resins, and various organic substances, including minerals, amino acids, flavonoids, polyphenols, ethanol, and different vitamins. It actively acts against hive-invading pathogenic microorganisms, including viruses and bacteria [14,15]. Finally, bee products are well known for their antiviral properties (Figure 1), which can surpass the standard available drugs in different cases. Indeed, propolis and honey have been observed to induce significant antiviral activities against different pathogenic mammalian viruses, including herpesviruses [16], influenza viruses [17], HIV, respiratory syncytial virus (RSV) [18], human T-cell leukemia virus type 1 (HLTV-1) [19], dengue virus (DENV) [20], Newcastle disease virus (NDV) [21], and poliovirus (PV). In addition, bee pollen and bee bread were efficacious antivirals against herpes virus types 1 and 2 (HSV-1 and HSV-2) in an in vitro study [22]. They were also reasonably effective against three influenza virus strains (H1N1, H3N2, and H5N1) [23]. Several clinical studies have suggested their practical use in place of antiviral drugs. Propolis can also be used to effectively inhibit HSV-1 and HSV-2 replication. A significant quantitative reduction in the viral copies of HSV-1 and HSV-2 confirmed the antiviral activity of propolis [24]. Vynograd et al. (2000) [25] investigated the efficacy of Canadian propolis ointment compared with acyclovir and placebo ointments in managing the herpes simplex virus. The healing capacity and the antimicrobial activity against the vaginal microflora were significantly enhanced by Canadian propolis as compared to acyclovir or placebo. Its antiviral activity was synergistically enhanced in combination with acyclovir [26]. Recently, bee products, more specifically propolis and honey, have been effectively clinically tested against SARS-CoV-2 due to their potential antiviral properties [27].

In addition to numerous antibacterial, antiviral, and anticancer activities, bee products have also generated significant antiprotozoal action against *Giardia lamblia* and Caenorhabditis [28,29,30]. Propolis was found to exert antimalarial effects against *Plasmodium* species, including *P. falciparum*, *P. ovale*, *P. vivax*, and *P. malariae* [31]. Several studies reported that royal jelly yielded antimicrobial activity against numerous pathogenic microorganisms [27]. Beeswax is another kind of bee product found to induce significant antimicrobial activity against pathogenic microbes [32]. Bee pollen has been reported to induce bactericidal and bacteriostatic activities [33]. The current review discusses the general composition and potential antiviral activities of bee products as a beneficial resource material for researchers to study the currently available studies and to discover more antiviral possibilities of bee products against mammalian viruses.

## 2. Honey

Honey produced by honeybees is a natural product extensively used as a food and medicinal source [34]. It is a complete food with several medicinal properties and has been used to treat gastrointestinal diseases, burns, wounds, inflammation, different ulcers, and abscesses and as a contraceptive [35].

### 2.1. Composition of Honey

Honey’s general composition is represented by different simple sugars and water. Its minor components, such as proteins and amino acids, enzymes, organic acids, flavonoids, volatile compounds, vitamins, and minerals, differentiate honeys originating from different botanical sources. Honey characteristics such as aroma, color, and flavor directly depend upon factors such as the honeybee species, floral source, climatic conditions, processing methods, packaging, and storage conditions [36]. Sugars are known as the major constituents of honey, ranging from 70–80%, containing about 70% monosaccharides (glucose and fructose) and about 10% di- and trisaccharides (maltose, sucrose, trehalose, turanose, isomaltose, kojibiose, maltulose, nigerose, maltotriose, erlose, and melezitose) [37,38]. In addition, proteins are also minor constituents of honey, including different enzymes and free amino acids. Among amino acids, proline is found in the highest amounts in honey. Furthermore, histidine, aspartic acid, glycine, glutamic acid, glutamine, threonine, tyrosine, alanine, a-alanine, b-alanine, aminobutyric acid, tryptophan, threonine, valine, arginine, methionine, lysine, serine, cysteine, asparagine, leucine, isoleucine, ornithine, and phenylalanine are also present in honey [39].

Honey contains different water-soluble vitamins, such as complex B (B1, B2, B3, B5, B6, B8, B9, and H) and vitamin C. Vitamin C is partly responsible for the antioxidant effect of honey [40]. Organic acids found in honey range from around 0.57% of the total weight and can induce electric conductivity and minor acidity, influencing the taste and color of honey. Organic acids are developed as a result of nectar conversion into honey following physical and chemical modifications. Some commonly found organic acids are citric acid, glutamic acid, aspartic acid, lactic acid, fumaric acid, butyric acid, malonic acid, acetic acid, gluconic acid, quinic acid, tartaric acid, formic acid, galacturonic acid, glyoxylic acid, malic acid, glutaric acid, propionic acid, α-hydroxyglutaric acid, lactic acid, 2-hydroxybutyric acid, pyruvic acid, oxalic acid, isocitric acid, succinic acid, α-ketoglutaric acid, 2-oxopentanoic acid, methylmalonic acid, shikimic acid, and gluconic acid, which is the most vital type of organic acid in the composition of honey [41]. More than 400 different volatile components, including benzene derivatives, C13-norisoprenoids, sesquiterpenes, monoterpenes, terpenes, fatty acids, alcohols, ketones, aldehydes, and esters, have been observed in honey. These compounds are responsible for their taste and may differ depending upon the nectar origin and processing and storage conditions [42,43]. The general composition of honey is given in Figure 2.

The phenolic components of honey are divided into flavonoids (anthocyanidin, chalcones, flavanols, flavones, flavanones, and isoflavones) and non-flavonoids (phenolic acids) (Figure 3) [44]. Non-flavonoids are present in two primary forms: hydroxycinnamic and hydroxybenzoic acids. Several flavonoids have been identified in honey, such as quercetin, myricetin, kaempferol, chrysin, pinocembrin, pinobanksin, galangin, hesperetin, caffeic acid, ellagic acid, syringic acid, ferulic acid, vanillic acid, *p*-coumaric acid, gallic acid, rosmarinic acid, benzoic acid, and chlorogenic acid. Some minor elements, including silver, arsenic, barium, calcium, cadmium, cobalt, chromium, copper, iron, potassium, iodine, magnesium, manganese, sodium, nickel, phosphorus, lithium, selenium, and zinc, are also found in honey [45]. The mineral content of honey ranges between 0.04% in light types and 0.2% in dark types of honey, respectively. Few heavy metals, in from 0.2–0.04% in dark and light kinds of honey, respectively. A few heavy metals, including arsenic, cadmium, mercury, and lead, are also observed in honey but should not surpass the maximum residual limit [46].

All of these components are of important nutritional value, with honey being a food product with valuable nutritional properties.

### 2.2. Antiviral Activity of Honey

Viral infections are considered the most prevalent and deadly infections among all other microbial infections because many viruses remain infectious for a long period in dry mucous [47]. As viruses replicate in the host cells, their destruction also means the destruction of host cells, and vaccination is the best method to prevent viral infections [48]. Several investigations have determined the therapeutic impact of honey and its bioactive compounds against many viral infections. They may protect against the respiratory syncytial virus, varicella-zoster virus (VZV), influenza, herpes simplex viruses, immunodeficiency virus, rubella virus, AIDS, rhinoconjunctivitis, gingivostomatitis, viral hepatitis, rabies, and SARS-CoV-2 (Table 1) [49,50,51,52,53,54,55,56,57,58,59]. The exact mechanism of action of honey and its bioactive compounds is immense and still unknown. Bioactive components such as quercetin, chrysin, kaempferol, etc., may provide antiviral activity by preventing the entry, entrapment, and replication of viruses (Figure 4).

Honey can be considered a significant strategy to treat VZV infection, as it can be easily applied to the skin. A study showed the safe and effective topical application of honey for the treatment of genital and herpes lesions [60,61]. An investigation of the antiviral activity of honey (clover and manuka honey) exhibited significant activity against VZV [62]. The antiviral properties of honey and its particular bioactive components have been analyzed against the respiratory syncytial virus (RSV). The obtained results indicated that treatment with honey encouraged viral replication inhibition. It has also been observed that sugar components are involved in virus inhibition, and simultaneously, it has also been found that methylglyoxal may enhance the antiviral potential of honey against RSV [18]. Based on the obtained results, it can be concluded that honey can be used as an alternative therapy for treating RSV. Influenza is highly contagious and more threatening than RSV in all age groups [63]. Influenza viruses mainly spread via inhalation transmission through sneezing or coughing droplets. Several studies have investigated the anti-influenza activity of honey [17]. Moreover, the significant synergistic effect of honey in combination with propolis has also been examined against the influenza virus [64].

### 2.3. Adverse Effects of Honey

Several toxic components have been identified in honey, such as diterpene grayanotoxins in honey from *Rhododendron ponticum* and *Rhododendron luteum* plants species of the rhododendron [65]. This type of honey is known as “mad honey”; it may cause serious neural intoxication, particularly in Turkey’s Black Sea area. Despite the mad honey’s toxicity, it is employed in folk medicine, especially for sexual dysfunction, hypertension, and some other illnesses [66,67].

*Fabaceae*, *Boraginaceae*, and *Asteraceae* plants generate pyrrolizidine alkaloid substances that are not toxic but can be transformed into harmful substances (pyrrolic metabolites) via the liver after the ingestion of honey [68]. Intoxication cases (characterized by memory loss, delirium, and seizures) have been reported in New Zealand, caused by consuming honey contaminated with neurotoxic sesquiterpene lactones hyenanchin and tutin [69,70,71]. Some other secondary metabolites of plants have been found in honey that are known to have harmful influences on humans, such as oleandrigenin and oleandrin (*Apocynaceae*), gelsemine and strychnine (*Gelsemiaceae*), saponins (*Sapindaceae*), hyoscine, and hyoscyamine (*Solanaceae*) [69,70,71,72].

## 3. Bee Pollen

Bee pollen is a mixture of flower pollen, nectar, and bee secretions, which honeybees transform into bee bread inside the hive, providing a nutritional source for colony formation and maintenance [73]. The insects gather flower pollen grains and mix them with nectar and/or honey and salivary secretions to amalgamate them, developing pollen loads. These formed grains are carried on their legs into the hive for storage, fermentation, and further consumption [74]. About 250 components have been identified in bee pollen [75]. Among these constituents, proteins, fatty acids, vitamins, and minerals (Figure 5) have been found to exert antibacterial, antioxidant, antifungal, antiallergic, anti-inflammatory, antitumor, and hepatoprotective activities [76]. The nutritive value of bee pollen is attributable to sugars, proteins, and lipids, which are found in balanced proportions in its composition [77,78].

### 3.1. Composition of Bee Pollen

The composition of bee pollen (Figure 5) varies greatly depending on the geographical and botanical source of the flower pollen grains, the main source of the developed pollen loads. Its main constituents include 5–60% proteins, 13–55% sugars, 0.3–20 crude fiber, and 4–7% lipids, amino acids, and nucleic acids [78]. The protein content in bee pollen may vary depending upon its botanical source, as presented in Table 2 [79,80,81,82,83,84,85,86,87].

Mineral components, including Ca, Cu, Fe, K, Mg, and Na, have been found in bee pollen. Several enzymes and vitamins, such as vitamin E (tocopherol), thiamine, folic acid, provitamin A (β-carotene), biotin, and niacin, have also been identified [88]. The fatty acid components include archaic, γ-linoleic, and linoleic acids, phytosterols, terpenes, and phospholipids [89]. Frequently found flavonoids are quercetin, catechins, isorhamnetin, and kaempferol [90]. Lutein was found to be the major carotenoid in *Callendula officinalis*, *Taraxacum officinale*, and *Anthylis* sp. bee pollen from Romania [91], as well as small amounts of β-criptoxanthin and β-carotene.

### 3.2. Antiviral Activity of Bee Pollen

Bee pollen has been documented as a significant nutritional supplement with several therapeutic actions, such as antibacterial, antifungal, antiallergic, antioxidant, anticancer, immunomodulatory, and hepatoprotective activities [76]. Only one study on its antiviral activity was found. Lee et al. (2016) investigated the anti-influenza activity by isolating a few bioactive components (one alkaloid and six flavonoids) of bee pollen. All of the bioactive components showed significant anti-influenza activity. The most effective results were observed with luteolin [23].

### 3.3. Adverse Effects of Bee Pollen

Bee pollen derived from different sources, such as *Echium vulgare*, *Senecio jacobaea*, and *Symphytum officinale*, may have hepatotoxic properties due to the presence of pyrrolizidine alkaloids at toxic levels [92]. Health issues associated with bee pollen consumption may arise from several other contaminants, such as bacteria, pesticides, heavy metals, and mycotoxins [93,94].

## 4. Bee Bread

Bee bread is formed by anaerobic fermentation by different *Lactobacillus* spp. of bee pollen in the hive. It contains different proteins, lipids, carbohydrates, fatty acids, free amino acids, water, vitamins, and some bioactive compounds [95]. It is used as a meaningful food supplement due to its high nutritional content. Its significance as a nutraceutical or food supplement is largely based on its composition [96].

### 4.1. Composition of Bee Bread

Bee bread (Figure 5) is prepared by worker bees using pollen collected from plants [97]. It is composed of proteins, lipids, carbohydrates, fatty acids, free amino acids, vitamins, minerals, water, and some bioactive components [98]. The protein content ranges from 14.1–37.3 g/100 g, with a mean value of about 23.1 g/100 g, quite similar to that of bee pollen (23.8 g/100 g). Some commonly found enzymes in bee bread are glucose-oxidase, phosphatase, and amylase. Bee bread contains several amino acids, including proline, glutamic acid, valine, aspartic acid, arginine, isoleucine, histidine, lysine, leucine, phenylalanine, methionine, tyrosine, tryptophan, alanine, serine, cysteine, and glycine.

The lipid content in bee bread composition varies depending upon the pollen-producing plant origin. Icosa-tetraenoic and octadecenoic acids, accounting for about 15%, are abundantly found unsaturated fatty acids [99]. The carbohydrate content of bee bread ranges from 24.40 to 34.80%. Fructose is present in the highest quantity, around 57.51%, followed by glucose, maltose, and sucrose at 42.59%, 3.34%, and 0.12% of the fresh weight, respectively. Minerals found in bee bread include Ca, P, Na, Mg, Mn, Fe, Cu, K, S, and Al. K is the mineral with the largest concentration content in bee bread, about 0.74%, followed by 0.65% P and Ca [89]. Nutritional value is very important in bee bread, as reported by several studies, due to the equilibrated composition of sugars, proteins, and lipids and also the presence of unsaturated fatty acids and essential amino acids [77,79,80,82,86,91,98].

### 4.2. Antiviral Activity of Bee Bread 

A few investigations have evaluated the microbicidal activity of bee bread against several bacterial and fungal pathogens [100,101]. However, as far as we know, the antiviral activity of bee bread has not been evaluated so far. Only one study was found on the antiviral activity of bee bread by Didaras et al. [102], who investigated the antiviral effect of bee bread and bee pollen against Enterovirus-D68. Bee bread and bee pollen showed a significant virucidal impact, obtained with IC_50_ values ranging from 0.048 and 5.45 mg/mL, respectively. This study suggested that bee bread and bee pollen are auspicious antiviral agents and should be further investigated against various viruses to scrutinize their antiviral potentiality.

## 5. Propolis

Propolis is a resinous bee product produced by different plants and collected by *Apis mellifera* bees with a waxy texture. Bees use their mandibles for the collection of plant resins to produce propolis. Chemically, it can be described as a complex mixture constituting several bioactive compounds with antifungal, antibacterial, antiparasitic, antiviral, immunomodulatory, and hepatoprotective actions [103].

### 5.1. Composition of Propolis

The composition of propolis (Figure 6) varies depending upon the floral source, collection time, and genetic background of bees [104]. Its content typically includes plant resins and balsams (55%), waxes (30%), aromatic and essential oils (10%), pollens (5%), and some other substances (5%) [105]. Moreover, more than 500 molecules have been recognized in propolis composition, including phenolics, wax, terpenes, sugars, proteins, vitamins, and amino acids [106]. The phenolic components comprise high amounts and different types of chemical constituents, including phenolic acids, flavonoids, aldehydes, coumarins, esters, simple phenols, and lignans. Flavonoids are distinct components of propolis, such as galangin, pinocembrin, chrysin, and pinobanksin. So far, more than 150 flavonoids have been identified and studied in propolis [105].

Flavanones, another substantial group of flavonoids, have also been identified in propolis. Forty different types of flavanones have been identified in propolis. Pinostrobin, liquiritigenin, pinocembrin, naringenin, isosakuranetin, and sakuranetin are the most commonly reported flavanones [107]. In addition to flavanones, some flavonols, and flavones such as quercetin, chrysin, acacetin, pectolinarigenin, tectochrysin, kaempferol, apigenin, galangin, fisetin, and izalpinin have also been identified [108].

Two major types of phenolic acids present in propolis are hydroxycinnamic acids (caffeic, ferulic, and *p*-coumaric acids) and hydroxybenzoic acids (salicylic gallic, protocatechuic, vanillic, and gentisic acids). The biological activities (pharmacological effects on living organisms) of propolis are directly dependent upon its composition (chemical). In ancient times, the Egyptians, Romans, and Greeks used propolis as a disinfectant and wound-healing substance. It has been reported as a safe (non-toxic) product for human use [109].

### 5.2. Antiviral Activity of Propolis

During the last few decades, several researchers have investigated the antiviral activities of propolis against various RNA and DNA viruses, such as HSV-1 and -2, PV-2, VSV, adenovirus-2, and an acyclovir-resistant mutant. Several studies have reported on the antiviral properties of different types of propolis and its bioactive components against various viruses, such as NDV, adenovirus, VSV, HSV, PV, influenza viruses, Vaccinia virus, VZV, IBDV, rotavirus, and coronavirus, as presented in Table 3 [26,110,111,112,113,114,115,116]. The antiviral activity of flavonols was observed to be more significant as compared to flavones present in propolis. The antiviral activity of flavone in combination with flavonol was also explored against HSV-1. The obtained results indicated a considerable synergistic effect that significantly increased the antiviral potential of propolis [117]. The inhibition of PV proliferation was examined by virus multistep replication and a plaque reduction assay. HSV titer was reduced to 1000 by propolis at a 30 µg/mL concentration, while adenovirus and VSV were observed to be less prone. The antiviral activity of propolis and, more specifically, its bioactive components, such as chrysin, quercetin, galangin, luteolin, kaempferol, and apigenin, were examined against HSV [118].

The antiviral potential of ethanol and aqueous propolis extracts and its components, such as chrysin, pinocembrin, galangin, and caffeic, benzoic, and *p*-coumaric acids, were analyzed against HSV-1, resulting in a more than 98% reduction in plaque formation [111]. The antiviral activity of caffeic acid was observed to be more or less sensitive, in that adeno and vaccinia viruses are more susceptible as compared to parainfluenza and poliovirus and less susceptible to influenza virus [119]. Ethanol and aqueous propolis extracts were investigated against HSV-1 and -2. These propolis extracts showed a 49% viral reduction against HSV-2 infectivity [120]. Thirteen different ethanol extracts were prepared using Brazilian green propolis and tested against the influenza virus. All 13 extracts showed significant in vivo and in vitro anti-influenza activity [53].

Propolis exhibited virucidal effects against enveloped viruses such as VSV and HSV. Propolis samples from four countries, namely, Germany, Egypt, Austria, and France, were studied against infectious bursal disease virus and avian reovirus (ARV). The obtained results revealed that all samples of propolis significantly decreased the infectivity of viruses [121]. Egyptian propolis exhibited the highest antiviral efficacy against IBDV and ARV [122]. Hydromethanolic geopropolis (HMG) extract was analyzed against the herpes virus via electron microscopy and viral DNA quantification and displayed 98% viral reduction [123]. The antiviral potential of Hatay (Turkish) propolis was evaluated against HSV-1 and -2. All concentrations (25, 50, and 100 µg/mL) of propolis showed significant antiviral potential against herpes viruses as compared to acyclovir. Moreover, a solid synergistic antiviral potential was observed when acyclovir was used in combination with propolis as compared to alone [124].

### 5.3. Adverse Effects of Propolis

Toxic effects of propolis have been observed at high doses, such as 15 g/day [125]. Despite the significant nutritional profile of propolis, it may induce allergic reactions. It has also been found that dermatitis patients are sensitive to propolis [126].

## 6. Bee Venom

Bee venom is a transparent and inodorous liquid amalgam of proteins (with 4.5–5.5 pH), mostly used by bees to protect themselves against different predators. In liquid form, one drop of it contains only 0.1 µg of venom (dried weight) and about 88% water. It is a complex mixture of different peptides, enzymes, minerals, and amino acids [127].

### 6.1. Composition of Bee Venom

Bee venom is prepared by female worker bees and constitutes several active compounds, including peptides such as apamin, melittin, adolapin, and mast cell degranulating peptide (MCD); enzymes such as hyaluronidase and phospholipase A2 (PLA2); minerals (Ca, Mg, and P); bioactive amines such as histamine, dopamine, and noradrenaline; and some volatile compounds containing complex ethers (Table 4) [128,129]. Melittin is known as its vital component, ranging from 40–60% of the entire composition [130]. Apamin, containing 2-disulfide bridges, is a polypeptide of 18 amino acids and is the smallest neurotoxin present in bee venom. Apamin can cross the blood–brain barrier (BBB) and hence can influence the functioning of the central nervous system [131]. MCD is very similar to the apamin peptide structure, as it also contains 2-disulfide bonds but with 22 amino acids. It is recorded at about 2–3% of the total dry weight of bee venom. Adolapin is about 0.5–1% of the total dry weight of bee venom and contains 103 amino acids. Phospholipase A2 is a polypeptide with four disulfide bridges containing 128 amino acids, also known as the lethal enzyme of bee venom. Hyaluronidase is another enzyme in bee venom composition with 1.5–2% dry weight and can break down hyaluronic acid present in tissues, such as in synovial bursa in patients with rheumatoid arthritis [132].

Various studies have reported the positive impacts of bee venom, providing antimicrobial, antiviral, anticancer, anti-inflammatory, antimalarial, and hemolytic effects based on the bioactive compounds of bee venom. The therapeutic efficacy of its bioactive compounds has been tested in the treatment of central nervous system diseases (Alzheimer’s disease and Parkinson’s disease), human inflammatory diseases, prostate and ovarian cancers, amyotrophic lateral sclerosis (ALS), HIV, and several other conditions [133,134,135].

### 6.2. Antiviral Activity of Bee Venom

Bee venom and its components have exhibited significant antiviral activities against several viruses (Table 5) [136,137,138,139], including herpes simplex virus, respiratory syncytial virus, vesicular stomatitis virus, influenza virus, enterovirus-71, influenza A virus, coxsackievirus, and papillomaviruses (HPVs). Papillomaviruses are known as the most common cause of cervical carcinoma induction. Bee venom can significantly inhibit cancer cell growth by downregulating E6/E7 proteins [136,140].

Melittin, a bee venom peptide, can activate the immune system against porcine reproductive and respiratory syndrome viruses (PRRSV) by upregulating Th1 cytokines (IL-12 and IFN-) and other immune cells, such as gd-T cells, CD3^+^–CD8^+^, and CD4^+^–CD8^+^, causing a decrease in viral load and a reduction in infection severity (in pigs infected with PRRSV) [137]. Phospholipase A2 and its constituent P3bv peptide exhibited considerable activity against human immunodeficiency virus (HIV) by preventing cellular fusion and inhibiting the virus’s (T-tropic) replication. Comparatively, PLA2 can inhibit T- and M-tropic viruses but cannot prevent cellular fusion. It has also been demonstrated that the peptides PLA2 and P3bv have both shown different inhibition mechanisms against HIV replication. P3bv is probably associated with a chemokine receptor, CXCR4, and simultaneously, PLA2 is associated with a binding receptor with high affinity [141]. Secreted phospholipase A2 (sPLA2) has shown significant antiviral potential against hepatitis C virus (HCV), Japanese encephalitis virus (JEV), and DENV, with IC_50_ 117 ± 43, 49 ± 13, and 183 ± 38 ng/mL, respectively [138].

## 7. Royal Jelly

Royal jelly (RJ) is a substance enriched with lipids and proteins developed by mandibular and hypopharyngeal bee glands [142]. This bee product is creamy and viscous in nature, with acidic pH of 3.1–3.9 and strong buffering capability (ranging from 4–7). The taste and odor of RJ are slightly bitter and partially water-soluble, having a 1.1 g/mL density [143].

### 7.1. Composition of Royal Jelly

Royal jelly (Figure 7) is an acidic colloidal substance containing water, carbohydrates, lipids, proteins, minerals, and vitamins [144]. Water is the principal constituent, ranging from 60–70%, followed by 10–18% carbohydrates, 9–18% proteins, 4–8% lipids, and 0.8–3% minerals and vitamins [145]. Glucose and fructose are the most abundant carbohydrates of RJ, followed by sucrose and small amounts of oligosaccharides, including maltose, isomaltose, melibiose, gentiobiose, trehalose, raffinose, ribose, and erlose [146]. The majority of the protein content belongs to apalbumins or the major royal jelly protein (MRJP) family, constituting up to 83–90% of the protein content [147]. The MRJP family contains nine identified proteins, MRJP1, MRJP2, MRJP3, MRJP4, MRJP5, MRJP6, MRJP7, MRJP8, and MRJP9, with 49–87 KDa molecular masses. Proteins other than MRJPs include apismin, apolipophorin III, glucose oxidase, jelleines, royalactina, and royalisin [148].

The lipid components are the unique feature of RJ composition. About 90% of lipids are composed of fatty acids, and the remaining components include waxes (5–6%), phenols (4–10%), steroids (3–4%), and phospholipids (0.4–0.8%) [143,149]. Several vitamins (water-soluble) have been identified in RJ composition. Vitamin B5 (52.80 mg/100 g) [150] is the chief vitamin component of the composition. Other vitamins found in its composition include vitamins B3, B6, E, B2, B1, C, A, B9, D, and B12 with particular amounts of 42.42, 11.90, 5.00, 2.77, 2.06, 2.00, 1.10, 0.40, 0.2, and 0.15 mg/100 g, respectively [151].

The mineral elements include Ca, Cu, Fe, K, Mn, Mg, Na, and Zn. Some trace elements have also been identified, including Cd, Al, Cr, Ba, Co, Bi, Mo, Ni, Sr, Sb, Sn, Hg, V, Pb, Te, Ti, and Tl [143]. Some minor compounds have also been identified, including heterocyclic components, neopterin, and biopterin. Other components are also present in low amounts, such as phosphatase, acetylcholine, free nucleotides (cytidine, adenosine, iridine, and cytidine), AMP, ADP, ATP, and benzoic, citric, lactic, and gluconic acids [144].

### 7.2. Antiviral Activity of Royal Jelly

Several investigations were found on its potential antibacterial and antifungal activities against various pathogenic microbes. Only a few studies have investigated its antiviral activities, and they showed significant antiviral activities against SARS-CoV-2 [152] and HSV-1 [56]. Another study revealed significant antiviral efficacy against the coxsackievirus by plaque assay [153].

### 7.3. Adverse Effects of Royal Jelly

Royal jelly can become contaminated with different environmental contaminants, such as carbamates, organochlorines, and organophosphorus (present in pesticides), which have less toxicity. Meanwhile, highly toxic chloramphenicol has also been identified [154]. Consumption of contaminated RJ and its proteins (MRJP-1 and MRJP-2) may induce anaphylaxis, dermatitis, and asthma [155].

## 8. Beeswax

Beeswax is an intricate compound produced in its liquified form in younger bees aged 12–18 days via particular wax glands. Its color changes from white to yellowish-brown following contact with bee honey and pollen. It partially dissolves in alcohol and completely dissolves in chloroform [32].

### 8.1. Composition of Beeswax

Beeswax (Figure 8) contains about 300 different components, such as 12–16% hydrocarbons (pentacosane, hentriacosane, heptacosane, nonacosane, and triacosane), 35–45% hydroxymonoesters and linear wax monoesters (basically derived from oleic, 15-hydroxypalmitic, and palmitic acids), 12–14% free fatty acids, 1% fatty alcohols, 15–27% esters and diesters of fatty acids, and some exogenous molecules (residues of pollen and propolis) [156]. Its composition may vary depending upon the different families and bee breeds [157]. Minerals (including Ca, Fe, Mn, P, Cu, K, Na, and Zn) and vitamins (including A, P, B6, B4, and B1) are also beeswax components. Beeswax is also used as an additive in the cosmetic, pharmaceutical, and food industries [158]. Moreover, it has also shown an antimicrobial effect against bacteria (*Staphylococcus aureus* and *Salmonella enterica*) and fungi (*Aspergillus niger* and *Candida albicans*) [159].

### 8.2. Antiviral Activity of Beeswax

Only one study evaluated the antiviral potential of four different beeswax extracts and the bee venom, alone and in combination, against DNA (Adeno-7 virus) and RNA (Rift valley fever virus) viruses [160]. According to the results of the study, acetone extract of black beeswax exhibited potent antiviral activity, with a depletion titer of 1.66 log (10)/mL. Ethanol extract of black beeswax also showed moderate activity, while acetone extract of black beeswax showed no antiviral activity.

## 9. Conclusions

Bee products, including honey, bee venom, propolis, royal jelly, bee bread, and beeswax, have high nutritional importance due to their chemical compositions. However, their consumption is conditioned by possible allergenic action. The increasing therapeutic potential of natural products, particularly bee products, has grabbed the attention of researchers over the last decades. Recent research developments and discernment of the biological properties of bee products and their bioactive compounds, responsible for generating antiviral, antibacterial, antifungal, anticancer, antiparasitic, hepatoprotective, and immunomodulatory effects, need to be extensively assimilated in an effort to upgrade the use of bee products for the management of viral infections and other diseases as well. Bee products contain several bioactive components that possess antiviral properties derived from plant sources. This review demonstrates the highly encouraging antiviral potential of bee products against various viruses. The antiviral components of bee products may often exert antimicrobial activities. Conclusively, this review also suggests that the consumption of bee products, i.e., honey, propolis, bee venom, bee pollen, royal jelly, bee bread, and beeswax, can be an integral approach to enhancing immunity and reducing human health problems.

## Figures and Tables

**Figure 1 nutrients-14-03579-f001:**
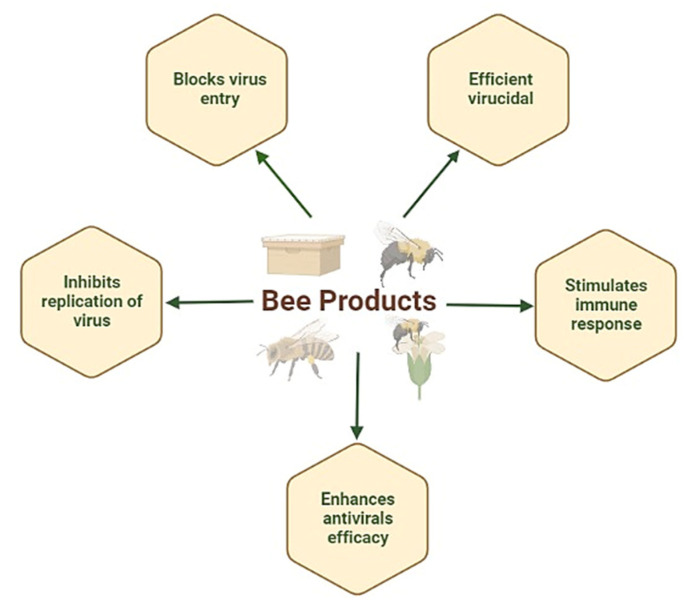
Antiviral properties of bee products.

**Figure 2 nutrients-14-03579-f002:**
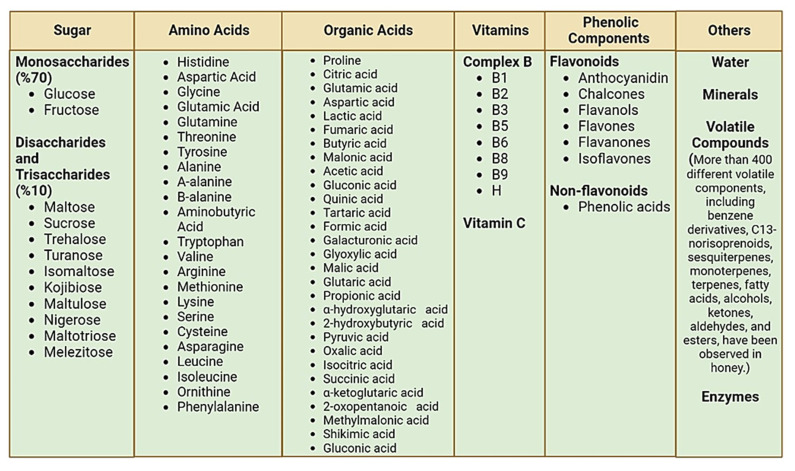
General composition of honey.

**Figure 3 nutrients-14-03579-f003:**
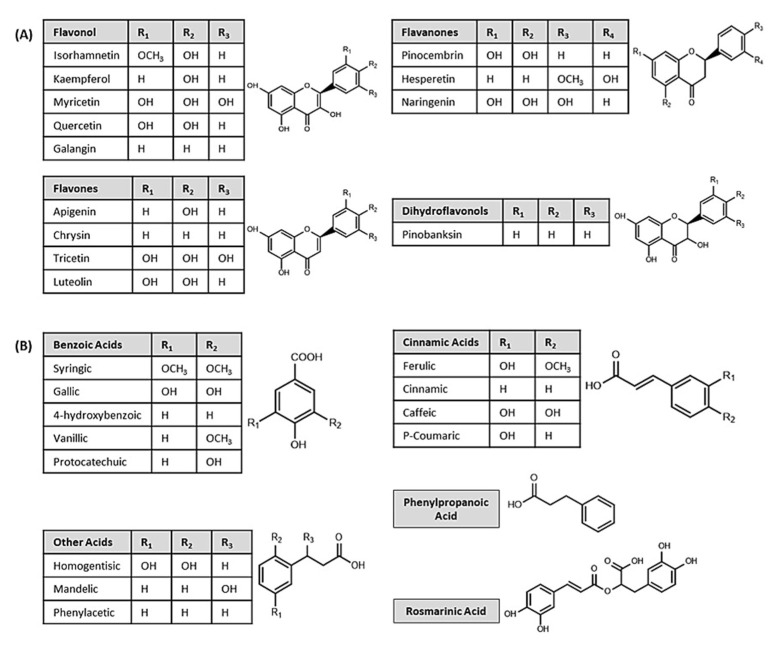
Chemical structures of some important phenolic components in honey; (**A**) flavonoids and (**B**) phenolic acids.

**Figure 4 nutrients-14-03579-f004:**
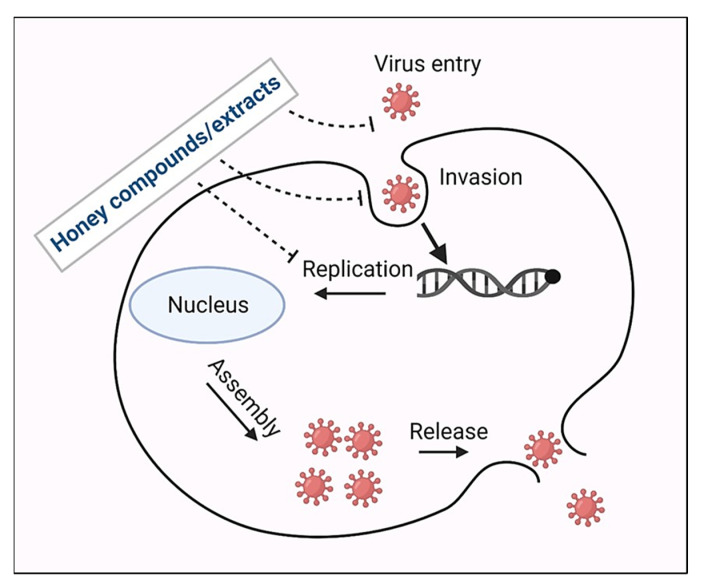
Honey and its components general mode of action against viruses.

**Figure 5 nutrients-14-03579-f005:**
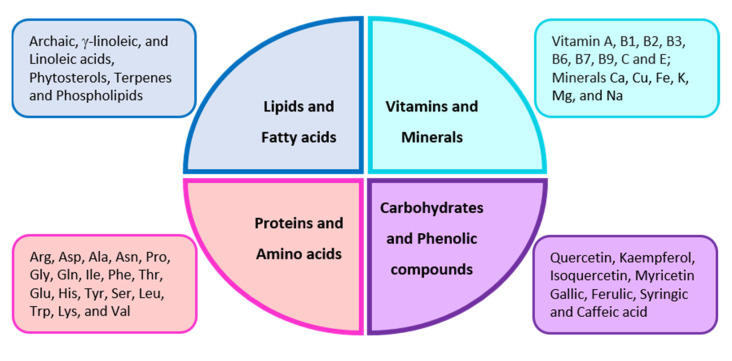
Main identified constituents of bee pollen and bee bread.

**Figure 6 nutrients-14-03579-f006:**
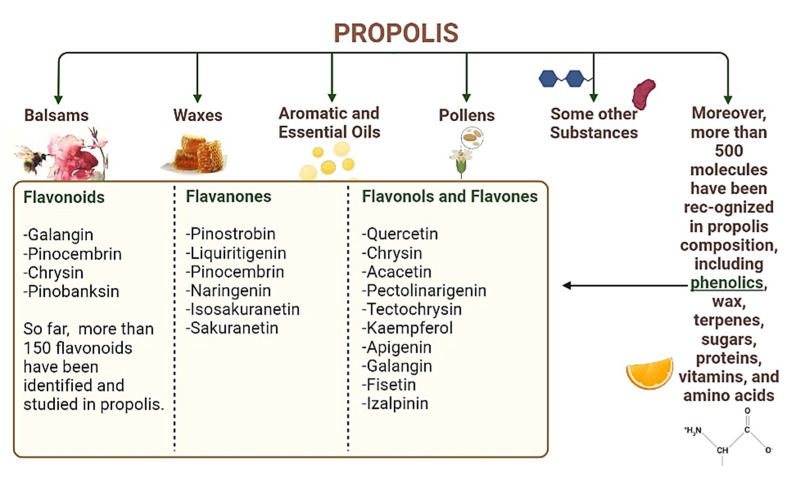
General composition of propolis.

**Figure 7 nutrients-14-03579-f007:**
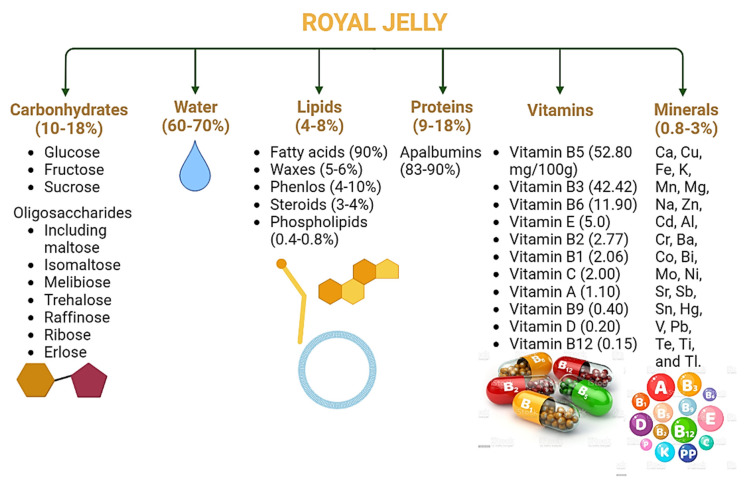
General composition of royal jelly.

**Figure 8 nutrients-14-03579-f008:**
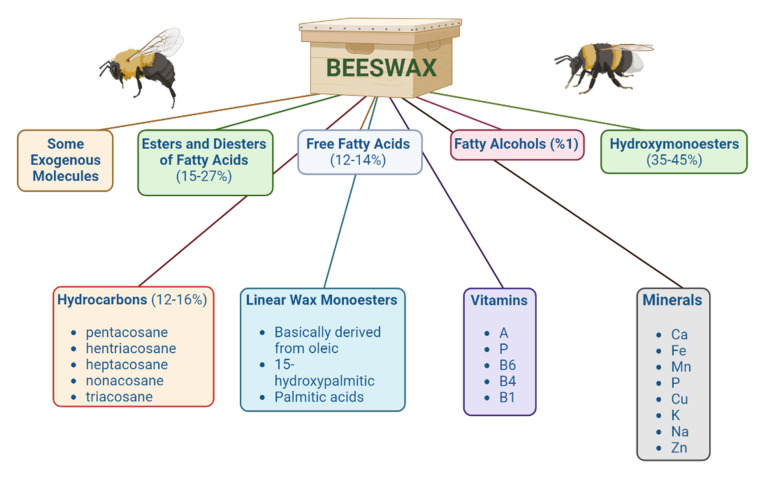
General composition of beeswax.

**Table 1 nutrients-14-03579-t001:** Antiviral activities of honey and its components.

Honey or Its Compounds	Organism	Tested Model	Outcome of the Study	References
Honey extract (Camelyn)	SARS-CoV-2	In vitro (plaque reduction assay) baby hamster kidney cell line 21 (BHK-21), bone marrow-derived hematopoietic stem cells (HSCs), and splenic cells	Showed an inhibitory effect with an EC_50_ value from 85.7 μg/mL to 192.4 μg/mL.	[49]
Honey	Rubella virus	In vitro African green monkey kidney cells	Inhibited the virus at all concentrations (1:1 to 1:1000 dilutions).	[50]
Honey	HSV-1	In vitro (plaque assay technique) Vero cells	Showed the highest inhibitory effect at 500 μg/mL and reduced viral load from 70795 to 43.3.	[51]
Methylglyoxal (Component of manuka honey)	Influenza B virus	In vitro Madin-Darby canine kidney cells	Inhibited influenza B virus replication, with 50% inhibitory concentrations ranging from 23–140 μM.	[52]
Kanuka honey	Herpes simplex labialis	Human model: 952 adults participated to compare the effect of Kanuka honey to 5% aciclovir cream	There was no difference in effectiveness compared with 5% acyclovir.	[53]
Honey	Common cold	Human model (cohort study): 122 students participated	Especially after six weeks of honey application, common cold frequency was lower than in the control group.	[54]
Tualang honey	HIV	Human model: 95 asymptomatic HIV-positive subjects participated	Ameliorated CD4 count, viral load, and quality of life.	[55]
Iranian honeys (8 monofloral honey types obtained from *Petro selinum sativum*, *Nigella sativa*, *Citrus sinensis*, *Zataria multiflora*, *Citrus aurantium, Zizyphus mauritiana*, *Astragalus gummifer*, and *Chamaemelum nobile* flowers)	HIV-1	In vitro peripheral blood mononuclear cells	Showed potent anti-HIV-1 activity in 6 of 8 monofloral honeys with EC_50_ values ranging from 5 to 105 µg/mL.	[56]
Honey	HSV-1	In vitro Vero cells	Showed complete inhibitory effect at 5% and higher concentrations.	[57]
Honey (garlic and ginger decoction)	Influenza virus	In vitro human peripheral blood mononuclear cells	Decreased replication of the H1N2.	[58]
Honeydew, manuka, and rewarewa honey	Adenovirus, rubella virus, and HSV	In vitro	Increased antiviral activity with the concentration of honey and time the virus was exposed to it.	[59]

**Table 2 nutrients-14-03579-t002:** Protein content of bee pollen originating from different countries.

Country	Origin	Protein Content (g/100 g)	References
Brazil	Heterofloral	8.4–40.5	[79]
	*Brassica napus*	23.0–24.5	[80]
	*Mimosa scabrella*	11.7–33.9	[81]
	*Mimosa caesalpiniaefolia*	17.6–21.2	
China	*Citrullus lanatus*	20.7	[82]
	*Fagopyrum esculentum*	14.3	
	*Helianthus annuus*	15.3	
	*Dendranthema indicum*	14.9	
Egypt	*Brassica kaber*	29.0	[83]
	*Zea mays*	23.3	
	*Trifolium alexandrium*	35.5	
Portugal	Heterofloral	18.8–34.2	[84]
	*Cistus*	23.0–27.1	[85]
Serbia	Heterofloral	14.8–27.2	[86]
	*Fabaceae*	19.9	
	*Salix*	24.8	
Spain	Heterofloral	12.5–20.8	[84,87]
	*Cistus*	12.6–22.5	

**Table 3 nutrients-14-03579-t003:** Antiviral activities of propolis and its components.

Propolis Type and Its Components	Organism	Tested Model	Outcome of the Study	References
Mexican propolis	Canine distemper virus	In vitro African green monkey kidney cells	Propolis application decreased viral expression and correlated with increased cell viability.	[110]
Propolis extracts	HSV-1	In vitro RC-37 cells	IC_50_ values of aqueous and ethanol extracts were determined at 0.0004% and 0.000035%, respectively.	[111]
Propolis	HIV-1	In vitro	Propolis abolished syncytium formation at 4.5 micrograms/mL and decreased p24 antigen production by as much as 90–100%.	[112]
Propolis extracts	HSV-1 and HSV-2	In vitro	Standardized preparations of propolis exhibited antiviral bioactivity.	[26]
Brazilian propolis (kaempferol, KF and p-coumaric acid, and p-CA)	Human rhinoviruses (HRVs)	In vitro HeLa cells	They inhibited HRV-3 infection when added during the early stages following virus inoculation.	[113]
Propolis	HSV-1	In vitro Vero cells in vivo newborn rats	The addition of 10% propolis extract led to 80–85% protection.	[114]
Propolis extract GH-2002	Varicella zoster virus	In vitro LEP cells	IC_50_ value was determined to be 64 μg/mL.	[115]
Propolis extract ACF^®^	HSV-1 and HSV-2	In vitro MDBK cells	Showed pronounced virucidal effect and interfered with virus adsorption.	[116]

**Table 4 nutrients-14-03579-t004:** Chemical components of bee venom.

Bee Venom Components		Dry Weight%
Peptides	Melittin	40–50
	Apamine, MCD	2–3
	Secapine	0.5–2
	Minimine	2
	Pamine	1–3
	Adolapine	0.5–1
	Protease inhibitor	0.1–0.8
	Procamine (A, B), tertiapine, cardiopep, and melittin-F	1–2
Proteins	Phospholipase A2	10–12
	Hyaluronidase	1–2
	Phosphatase and phospholipase B	1
	α-Glucosidase	0–6
Sugars	Glucose and fructose	2–4
Minerals	Ca, Mg, and P	3–4
Amines	Aminobutyric acid, α-amino acids	1
	Noradrenaline	0.1–0.5
	Histamine	0.5–2
	Dopamine	0.2–1
Volatile compounds (pheromones)	Complex ethers	4–8

**Table 5 nutrients-14-03579-t005:** Antiviral activities of bee venom and its components.

Bee Venom or Extract	Organism	Tested Model	Outcome of the Study	References
Bee venom	Vesicular stomatitis virus (VSV), coxsackievirus (H3), herpes simplex virus (HSV), enterovirus-71 (EV-71), influenza A virus (PR8), and respiratory syncytial virus (RSV)	In vitro HEK293T, MDCK, HEp2, Vero cells, and HeLa	It is concluded that bee venom would be a promising antiviral agent, especially in the establishment of a broad-spectrum antiviral agent.	[136]
Bee venom	Porcine reproductive and respiratory syndrome virus (PRRSV)	In vivo *pigs*	Especially nasal or rectal application of bee venom may be used in the prevention of this infection in pigs.	[137]
Bee venom (phospholipase A2)	Dengue virus (DENV), hepatitis C virus (HCV), and Japanese encephalitis virus (JEV)	In vitro Huh7it-1 cells, MDCK, HEK293T, and Vero cells and in vivo embryonated eggs	Phospholipase A2 and its derivatives could be potent candidates for the development of broad-spectrum antiviral drugs that exert their effects by targeting viral envelope lipid bilayers derived from the ER membrane.	[138]
Bee venom	Lumpy skin disease virus (LSDV)	In vitro Maiden-Darby bovine kidney cells (MDBK), Hep-2, and MCF7 and in vivo embryonated chicken eggs	Bee venom could serve as a good treatment for LSDV after determination of suitable therapeutic doses.	[139]

## Data Availability

Not applicable.

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
