# Peer review of "General Nutritional Profile of Bee Products and Their Potential Antiviral Properties against Mammalian Viruses"

_nutrients, 2022, doi:10.3390/nu14173579_

Round 1

Reviewer 1 Report

1.      The topic is very interesting. Unfortunately, this article does not provide full picture on the antiviral properties for the readers or the scientific community. Authors should focus on the main topic.

2.      Title should be more focused on the antiviral action of bee products.

3.      Line 43: delete “cancer also”

4.      Some references don't fit with its citation/text sections, please adjust.

5.      Lines 44-48: “Cancer is one of the…………. of cancers [3,6] ”, the statement seems not in place, please delete .

6.      Line 53” “biotoxin”, delete

7.      Lines 150-157: please support by references

8.      Tables 1, 3, and 5: add a column with the outcome of the study as well as the mode of action .

9.      Figure 1: better to add the word (bee products) instead of (products)

10.  p-coumaric” would be changed to “p-coumaric” , in-vivo and in-vitro” to “in-vivo and in-vitro

11.  Line 343: “Toxic effects of propolis have been observed at high doses such as 15 g/die [128].”, rephrase

12.  Lines 380-381: “Bee venom can significantly……………. E6/E7 proteins [139,140]”, delete

13.  The studies reported throughout the manuscript lack details and discussion of the results.

14.  The mechanism of action should be discussed.

15.  Rewrite the conclusion section.

16.  Figure 2: please adjust the structures.

17.  English editing is highly recommended.

18.  This reference would be of benefit” El-Seedi, H., et al., 2020. Antimicrobial properties of Apis mellifera’s bee venom. Toxins12(7), p.451.

Author Response

Thank you for the observations which improve the quality of the manuscript. Attached we have answer punctually to all raised questions

Reviewer 2 Report

In my opinion, this research article is interesting because it aims to provide an elaborated vision of the antiviral activities of bee products with recent advances in research. The findings from this study may support the potential of bee products and their bioactive components/extracts as an alternative strategy to alleviate human health from an individual to a communal level. However, there are a few suggestions for the authors to consider in revising their manuscript:

1.     What is the significance of explaining cancer and honey in detail in the introduction?

2.     Please recheck the statements in lines 63 and 64.

3.     Please summarize the composition of bee products (honey, bee bread, propolis, royal jelly and beeswax) in table form.

4.     All studies listed in the summary tables need to be elaborated on in the text.

5.     Line 343; 15g/die or day?

6.     Elaboration on the antiviral activity of royal jelly and beeswax is needed.

7.     Based on the adverse effects explained in this manuscript, what is your recommendation regarding the consumption of bee products?

8.     Please add figures to make this review article more attractive and get more visualization. For example, maybe a figure that summarizes all the antiviral activities of bee products.

Author Response

Thank you for your observations. With their help we improve the quality of our manuscript. Attached we answer punctually to all your questions.

Author Response

We thank the reviewer for his/her observations. Attached we have answered punctually to all questions.

Round 2

Reviewer 1 Report

Accept in the current form

Reviewer 2 Report

All comments have been addressed by the authors.